# Effects of Treated Manure Conditions on Ammonia and Hydrogen Sulfide Emissions from a Swine Finishing Barn Equipped with Semicontinuous Pit Recharge System in Summer

**Jisoo Wi [1], Seunghun Lee [1], Eunjong Kim [1], Myeongseong Lee [1], Jacek A. Koziel [2],\* and Heekwon Ahn [1],\***

[1] Department of Animal Biosystems Sciences, Chungnam National University, Daejeon 34134, Korea; jswi@cnu.ac.kr (J.W.), huny9261@cnu.ac.kr (S.L.), ejkim0128@cnu.ac.kr (E.K.), leefame@cnu.ac.kr (M.L.),

[2] Department of Agricultural and Biosystems Engineering, Iowa State University, Ames, IA 50011, USA

\* Correspondence: hkahn@cnu.ac.kr (H.A.); koziel@iastate.edu (J.K.);
   Tel.: +82-042-821-5785 (H.A.); +1-515-294-4206 (J.K.)

**Abstract:** Gaseous emissions from animal production systems affect the local and regional air quality. Proven farm-scale mitigation technologies are needed to lower these emissions and to provide management practices that are feasible and sustainable. In this research, we evaluate the performance of a unique approach that simultaneously mitigates emissions and improves air quality inside a barn equipped with a manure pit recharge system. Specifically, we tested the effects of summertime feeding rations (used by farmers to cope with animal heat stress) and manure management. To date, the pit recharge system has been proven to be effective in mitigating both ammonia ($NH_3$; approximately 53%) and hydrogen sulfide ($H_2S$; approximately 84%) emissions during mild climate conditions. However, its performance during the hot season with a high crude protein diet and high nitrogen loading into the pit manure recharge system is unknown. Therefore, we compared the emissions and indoor air quality of the rooms (240 pigs, ~80 kg each) equipped with a conventional slurry and pit recharge system. The main findings highlight the importance and impact of seasonal variation and diet and manure management practices. We observed 31% greater $NH_3$ emissions from the pit recharge system ($33.7 \pm 1.4$ g·head$^{-1}$·day$^{-1}$) compared with a conventional slurry system ($25.9 \pm 2.4$ g·head$^{-1}$·day$^{-1}$). Additionally, the $NH_3$ concentration inside the barn was higher (by 24%) in the pit recharge system compared with the control. On the other hand, $H_2S$ emissions were 55% lower in the pit recharge system ($628 \pm 47$ mg·head$^{-1}$·day$^{-1}$) compared with a conventional slurry pit ($1400 \pm 132$ mg·head$^{-1}$·day$^{-1}$). Additionally, the $H_2S$ concentration inside the barn was lower (by 54%) in the pit recharge system compared with the control. The characteristics of the pit recharge liquid (i.e., aerobically treated manure), such as the total nitrogen (TN) and ammonium N ($NH_4$-N) contents, contributed to the higher $NH_3$ emissions from the pit recharge system in summer. However, their influence on $H_2S$ emissions had a relatively low impact, i.e., emissions were still reduced, similarly as they were in mild climate conditions. Overall, it is necessary to consider a seasonal diet and manure management practices when evaluating emissions and indoor air quality. Further research on minimizing the seasonal nitrogen loading and optimizing pit recharge manure characteristics is warranted.

**Keywords:** emissions; air quality; ammonia; hydrogen sulfide; pit recharge system; waste management; high-crude protein diet; swine production; animal production systems; sustainability

## 1. Introduction

Emissions from livestock facilities are composed of various compounds, including ammonia ($NH_3$), hydrogen sulfide ($H_2S$), odorous volatile organic compounds (VOCs), and particular matter (PM). Due to negative effects on the environment and occupational hygiene, $NH_3$ and $H_2S$ are considered as some of the most important pollutants associated with livestock production [1]. Gaseous $NH_3$ released from animal manure to the atmosphere causes eutrophication of surface water and soil acidification, and reduces biodiversity [2–5]. $NH_3$ is also considered a significant contributor to the formation of $PM_{2.5}$ and aerosols that result in haze and health concerns [1,6–8]. The aerosols in swine confinement buildings can lead to respiratory discomfort in pigs and can contribute to the suppression of feed intake and growth [9–11]. $H_2S$ produced from anaerobic decomposition of animal manure has a strong odor, even at very low concentrations. $H_2S$ has been responsible for many deaths of humans and animals in livestock facilities [12,13]. $NH_3$ and $H_2S$ are correlated with odor [14] and are relatively easy to measure using real-time sensors. Therefore, $NH_3$ and $H_2S$ have been used as representative surrogate gases of livestock odor and indicators of air quality. Since North America began collecting ammonia data using swine house field monitoring technology in the 1980s, researchers have investigated $NH_3$ and $H_2S$ emissions in swine facilities under various conditions [1]. Faulkner and Shaw [15] estimated $NH_3$ emissions of pigs by growth stage (farrowing, nursery, finishing) and reported a composite factor of 5.8 kg·head$^{-1}$·year$^{-1}$. Harper et al. [16] investigated seasonal $NH_3$ emissions and reported that emissions in summer were 3.2 times higher than in winter. Additionally, the National Air Emissions Monitoring Study (NAEMS) monitored carbon dioxide, methane, volatile organic compounds, and particular matter ($PM_{10}$ and $PM_{2.5}$), as well as $NH_3$ and $H_2S$ generated in livestock facilities [17].

There are hundreds of swine farms using semicontinuous pit recharge systems to improve indoor air quality and reduce gas emissions in the Republic of Korea. Aerobically treated liquid manure with autothermal thermophilic aerobic digestion (ATAD) [18] is pumped back into the slurry pit. Because the treated liquid dilutes the raw manure, a reduction of gaseous emissions can be expected [19–21]. Wi et al. [22] reported the reduction of $NH_3$ and $H_2S$ emissions from finishing pig housing equipped with a semicontinuous pit recharge system in mild seasonal conditions by ~53 and ~84%, respectively.

However, the $NH_3$ and $H_2S$ gas mitigation performance depend on the quality of the recharging liquid. When manure is treated in the ATAD system, the organic N is decomposed and converted to ammoniacal N and stabilized in nitrate ($NO_3^-$) form. If the ammoniacal N is high in recharging liquid due to insufficient aeration, high N input (e.g., via high protein content diet), and elevated manure pH, this may increase the $NH_3$ emissions from the pit situated under the slatted barn floor. If the pH of the liquid increases from 7 to 9 (at 20 °C), the fraction of free $NH_3$ increases from 3.8% to 28.4% [23]. The operational conditions of the ATAD (e.g., manure temperature and hydraulic retention time, HRT) also influence the characteristics of the recharging liquid.

Variations in N content in excreted manure are influenced by the feeding program, i.e., a common practice is to adjust the protein in the feed based on the growing stage and needs [6,19,24,25]. Especially in summer, farmers will feed a high crude protein diet to overcome heat stress and improve weight gain [26,27]. This protein-rich feed induces more N excretion in manure [19], resulting in increased N loading into the ATAD system. This, in turn, overloads the capacity of the ATAD process and leads to the return of high $NH_4^+$-N in the recharging liquid to the slurry pit. In addition, a high crude protein diet is related to higher $NH_3$ emission from swine manure itself. It is generally known that an additional 1% of crude protein content could increase $NH_3$ emissions by ~20% [25].

Seasonal operational ATAD conditions affect the characteristics of the recharging liquid. The ATAD-treated manure ("liquid fertilizer") is stored in a large-volume on-farm storage tank and is then generally applied to the field in spring and fall. However, many farms do not have the capacity to store the treated manure. Therefore, farmers empty the treated liquid from the storage tank in spring (when the fertilizer is most needed). Farmers even pump out the liquid manure from the

ATAD, disrupting the microorganism balance in the system. The disrupted ATAD system produces high $NH_4^+$ and partially treated manure in the recharging liquid, and likely affects gas emissions from the swine room and indoor air quality, especially in the hot season that follows. However, the exact extent of the effects on emissions and indoor air quality is not known.

Thus, the main objective of this research was to evaluate the $NH_3$ and $H_2S$ emissions from a semicontinuous pit recharge system with partially aerobically treated liquid manure in summer. The $NH_3$ and $H_2S$ emissions and concentrations inside the barn associated with the conventional slurry pit and recharged pit were evaluated and then compared in the context of mild seasonal conditions.

## 2. Experiments

### 2.1. Farm Description and Experimental Design

The experimental site was a commercial pork production farm, which was the same barn as in the study by Wi et al. [22]. The experiment was carried out for two weeks in mid-summer (17th to 30th July, 2018), and the average outside temperature of the barn was $29.0 \pm 1.0$ °C. The farm included a swine confinement building with a semicontinuous pit recharge system, solid–liquid separation facility, and autothermal thermophilic aerobic digestion (ATAD) system to produce aerobically treated liquid manure (HRT: 30 days), which recharged into the pit. The semicontinuous pit recharge system from the swine farm is depicted in Figure 1.

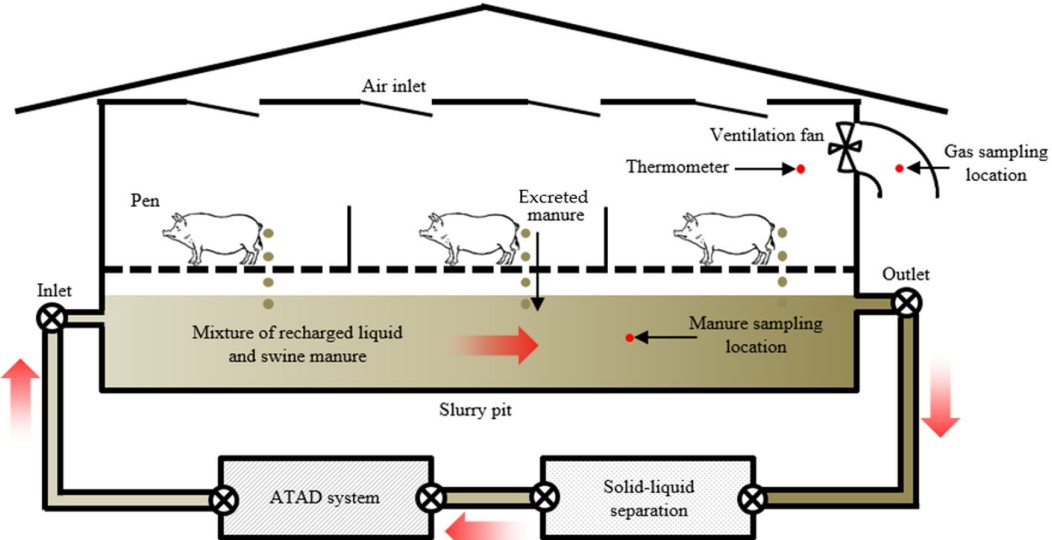

**Figure 1.** Schematic diagram of the facilities of the experimental farm with a semicontinuous pit recharge system. The excreted swine manure is circulated into the pit as a recharged liquid through the solid–liquid separation and autothermal thermophilic aerobic digestion (ATAD) system. Red arrows indicate the flow of the swine manure in a semicontinuous pit recharge system.

Two identical swine rooms in a building used for fattening swine were monitored as the control (conventional slurry pit; without pit recharge system) and ATAD treatment (with pit recharge system). For the ATAD treatment room, the pit was recharged with 12 $m^3$ of aerobically treated liquid manure daily, as described in a previous study [22]. The total pit volume of the ATAD treatment room was kept at a constant depth (about 80 cm) throughout the whole experiment period, and the depth of the manure in the control room was 82 cm on day 7. Manure from the control was emptied out every 2–3 months and carried to a centralized manure ATAD treatment plant.

The treated liquid manure from the pit recharge system is considered "liquid fertilizer" and is periodically pumped out to a large on-farm storage tank, then applied to the land before sowing and after harvest season (e.g., 2 times per year). In the case of the experimental barn, the farmer pumped

out the aerobically treated liquid manure from the storage tank and part of the ATAD system about two months before the experimental period.

Each room had 240 pigs weighing approximately 80 kg, and the stocking density was 0.79 m² head⁻¹. Among the three wall-mounted exhaust ventilation fans (Figures 1 and 2), one primary ventilation fan (Φ 550 mm) operated continuously at a constant rate (around 88 m³ min⁻¹), while the others (Φ 1000 mm) operated at variable speeds of 110–210 m³·min⁻¹ to maintain a set room temperature (25 °C), as described by Wi et al. [22]. During the whole experimental period for this study, two of Φ 1000-mm fans operated mainly in continuous mode. The gas sampling location of each room was directly downstream of each continuous operating fan. More details about the layout of the farm (schematic of the farm, top view of the swine room, etc.) are presented by Wi et al. [22] (Figures 1, A1, and A3).

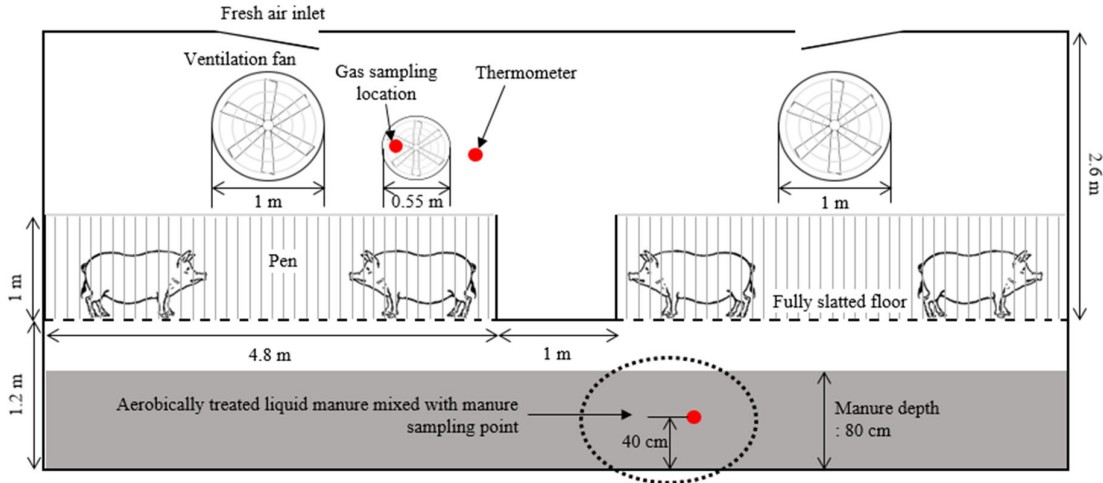

**Figure 2.** Side view of tested swine rooms (control and ATAD treatment). The gas sampling tube was positioned immediately downstream from a primary fan (operating at all times at constant rates). The manure was sampled from the middle of the pit.

## 2.2. Animal Lifecycle and Feeding Program

The experimental barn applied an all-in-all-out system; 240 growing pigs weighing around 30 kg were introduced into the growing–finishing barn; after 90 to 100 days the pigs are marketed, weighing ~115 kg. The pigs were fed two types of feed, depending on their weight (age) and the season (temperature). Due to the high ambient temperature in summer, the fattening period was 10–15 days longer than in fall or winter. Additionally, the feeding programs for the fattening period differed by season. In this research (summer), the pigs were fed with feed A (Table 1). However, in the case of a previous study [22], although the pig weights were similar to this research, the pigs were fed with feed B due to the relatively cool weather. The feeding program for the experimental barn is depicted in Figure 3. Additionally, Table 1 describes the characteristics of feeds A and B used for this research and compare them with the cool season diet [22], respectively.

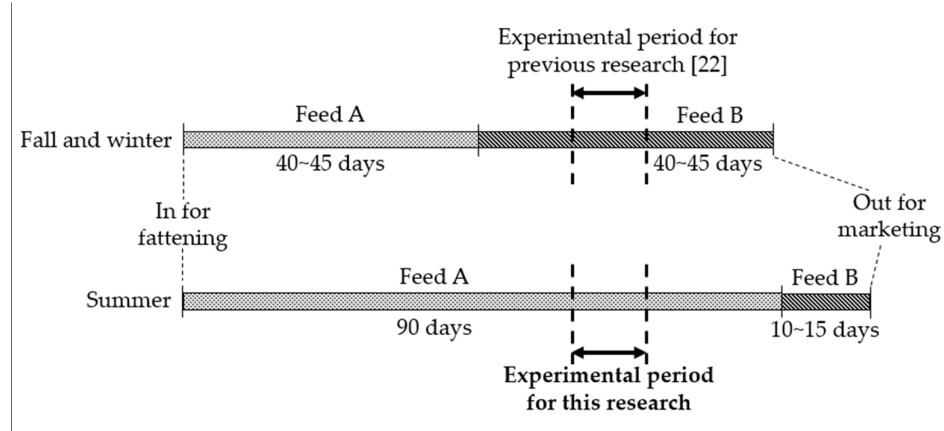

**Figure 3.** Comparison of 2 different feeding programs for pigs in the experimental barn. Due to this research being conducted in summer, 240 pigs (weighing ~80 kg) were fed with feed A.

**Table 1.** Characteristics of swine feed used in this study (summer) and the cool season (fall) diet.

| Item | Feed A (This Research) | Feed B (Previous Research, [22]) |
|---|---|---|
| Crude protein (%, d. b[1]) | 18.36 | 17.48 |
| Fat (%, d. b.) | 2.69 | 2.91 |
| Crude fiber (%, d. b.) | 5.16 | 5.75 |

Note: [1] dry basis.

### 2.3. $NH_3$ and $H_2S$

The identical real-time monitoring systems from the previous study [22] were used to measure $NH_3$ (NH3/CR-200) and $H_2S$ (H2S/C-50) concentrations and ventilation rates. The detailed performance data for both sensors are shown in Table 2. The $NH_3$ and $H_2S$ emissions were also estimated in the same way as described in detail by Wi et al. [22].

**Table 2.** Specifications of gas sensors (Membrapor, Co.) used in this study.

| | $NH_3$ | $H_2S$ |
|---|---|---|
| Model | NH3/CR-200 | H2S/C-50 |
| Detecting range | 0–100 ppm | 0–50 ppm |
| Resolution | 0.1 ppm | 50 ppb |
| Linearity ($R^2$) | 0.99 | 0.99 |

### 2.4. Manure and Feed Analysis

The recharging manure (aerobically treated) was sampled on day 7 of the experiment and recharged liquid mixed with manure was collected from the slurry pit under the swine room once. Manure from the conventional slurry pit (control) was also sampled on day 7. Samples from each pit were collected in the middle of the manure (40 cm from bottom) height (Figure 1). Manure samples were stored below 4 °C and analyzed for total solids (TS), volatile solids (VS), pH, electric conductivity (EC), total nitrogen (TN), and ammonium nitrogen ($NH_4$-N). TS and VS were assayed using the standard American Public Health Association (APHA) methods [28]. The pH and EC were measured with a digital pH meter with a combination glass electrode (Thermo Scientific, Orion 4 Star pH and EC conductivity benchtop meter). The TN content in manure was analyzed with the modified

Gunning method (using a sulfuric–salicylic acid mixture). The photometric analysis was used (Thermo Scientific, Gallery Discrete Analyzer) to detect $NH_4$-N in manure.

The feed for pigs was collected from the feed bin on day 7 and stored in the refrigerator, which was maintained below 4 °C. Then, feed samples were analyzed for crude protein (CP), fat, and crude fiber (CF) contents. The CP content was analyzed using the Kjeldahl method. The fat and CF contents in the feed were measured using the ether extract (EE) method and neutral detergent fiber (NDF) method, respectively.

### 2.5. Statistical Analysis

The values, including the concentrations and estimated emissions of $NH_3$ and $H_2S$ in two different rooms, were evaluated with Origin Pro software (Origin Lab, version 9) for statistical significance using a two-sample T-test. A significant difference between the control and ATAD treatment was determined at a significance level of $p < 0.05$.

## 3. Results

### 3.1. NH3 Emissions

The hourly trends of $NH_3$ concentrations and emission rates from control and ATAD treatment swine rooms are plotted in Figure 4. In both rooms, the distinct diurnal pattern of $NH_3$ concentration was repeated throughout the whole experimental period. The $NH_3$ concentration of the control room ranged from 10.5 to 19.1 ppmv. The range of $NH_3$ concentration of ATAD treatment ranged from 10.6 to 22.7 ppmv (Figure 4a), which was higher than the control room ($p < 0.05$) when comparing the average $NH_3$ concentrations for the control and ATAD treatment (Table 3).

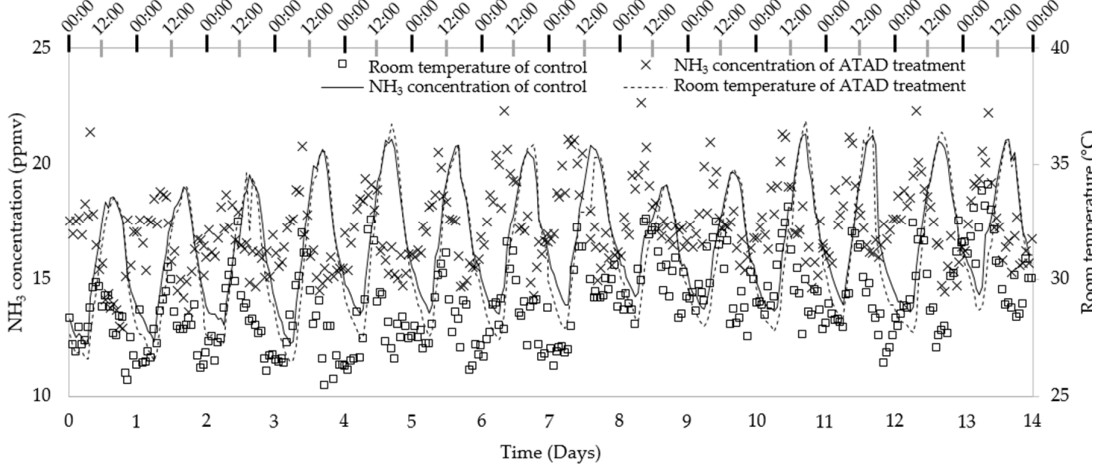

(a)

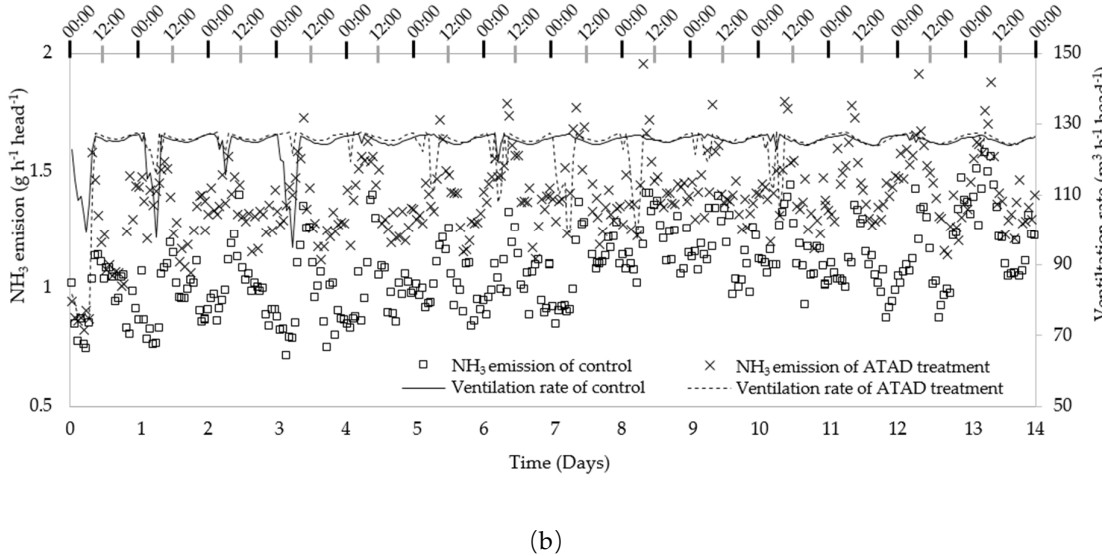

(b)

**Figure 4.** Comparison of hourly mean $NH_3$ concentrations and emissions from control (conventional slurry pit) and ATAD treatment (pit recharge system) in summer: (**a**) variation of measured concentrations and room temperatures for each room (control and ATAD treatment); (**b**) estimated emission and ventilation rates.

Except for the early mornings, which had relatively low outside temperatures, the ventilation rates for both rooms remained at approximately 125 $m^3 \cdot h^{-1} \cdot head^{-1}$ during most of the experimental period, which is the highest level possible in the ventilation system (Figure 4b). The ranges of $NH_3$ emission rates were 0.7–1.6 and 0.8–2.0 $g \cdot h^{-1} \cdot head^{-1}$ in control and ATAD treatments, respectively. The $NH_3$ emissions fully reflected changes in the concentrations for both rooms; the correlation coefficients (R) between $NH_3$ concentration and emission were 0.97 and 0.95 for the control and for the ATAD treatment, respectively (Figure 5).

Due to the slight variation in the ventilation rates, the correlation between the ventilation rate and $NH_3$ emission was poor in both rooms (R = 0.35 and 0.18 for control and ATAD treatment, respectively; Figure A1). However, when the ventilation rates were grouped into two levels (low and maximum), the observed correlations between the ventilation rates and $NH_3$ emissions at two levels showed a different trend. The maximum ventilation levels were determined by the operating rate of the ventilation fans, as measured by the airflow measurement assembly (AMA, [22]); the start points for the maximum ventilation level were 123 and 124 $m^3 \cdot h^{-1} \cdot head^{-1}$ for control and ATAD treatment, respectively (Figure A2). The ventilation rates for the control and ATAD treatment were maintained at the maximum level for 94% and 86% of the experimental period, respectively, while the average room temperature at the maximum ventilation level in both rooms was over 31.5 °C (Table A1). During the low ventilation levels, the average room temperature for both rooms was around 28.0 °C. At the low ventilation level, we observed a strong correlation between the ventilation rate and $NH_3$ emission in both rooms (R = 0.61 and 0.82 for control and ATAD treatment, respectively; Figure A2). On the other hand, at the maximum ventilation level, the correlations were relatively weak—the correlation coefficient (R) for the control was −0.02, while for ATAD treatment, it was 0.40 (Figure A2).

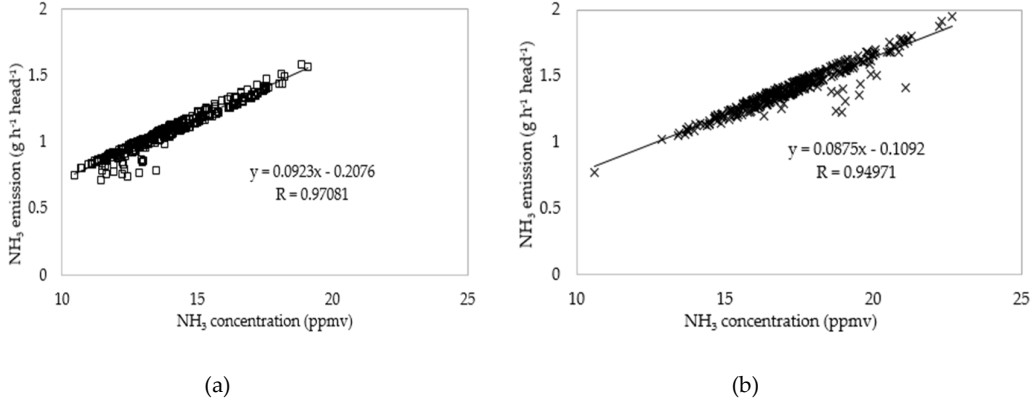

(a)
(b)

**Figure 5.** Correlation between $NH_3$ concentration and emission rates for the control (conventional slurry pit) and ATAD treatment (pit recharge system) in summer. (**a**) The correlation coefficient between $NH_3$ concentration and emission for the control (conventional slurry pit) was 0.97, (**b**) while for the ATAD treatment (pit recharge system) was 0.95.

*3.2. $H_2S$ Emissions*

The measured $H_2S$ concentrations for both rooms are shown in Figure 6a. During the experimental period, the levels of $H_2S$ concentrations and the emission rates in the ATAD treatment room were lower than the control. The range of $H_2S$ concentrations in the control room was 179–546 ppbv, while for ATAD treatment this range was 125–417 ppbv, showing a 54% overall reduction. Additionally, the ranges of $H_2S$ emissions were 26–88 and 16–64 mg·h$^{-1}$·head$^{-1}$ for the control and ATAD treatment, respectively, while the average $H_2S$ concentration for the control was statistically higher ($p < 0.05$, Table 3).

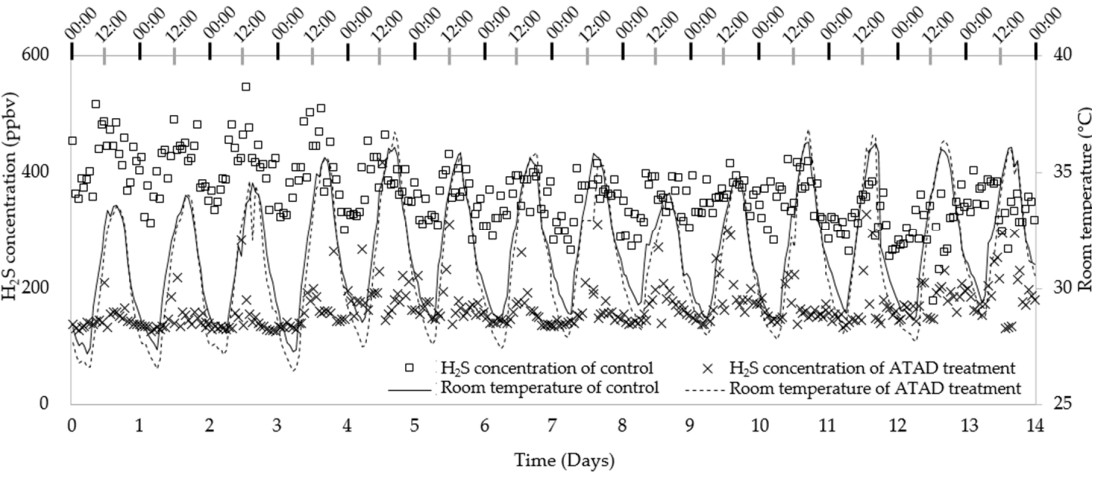

(a)

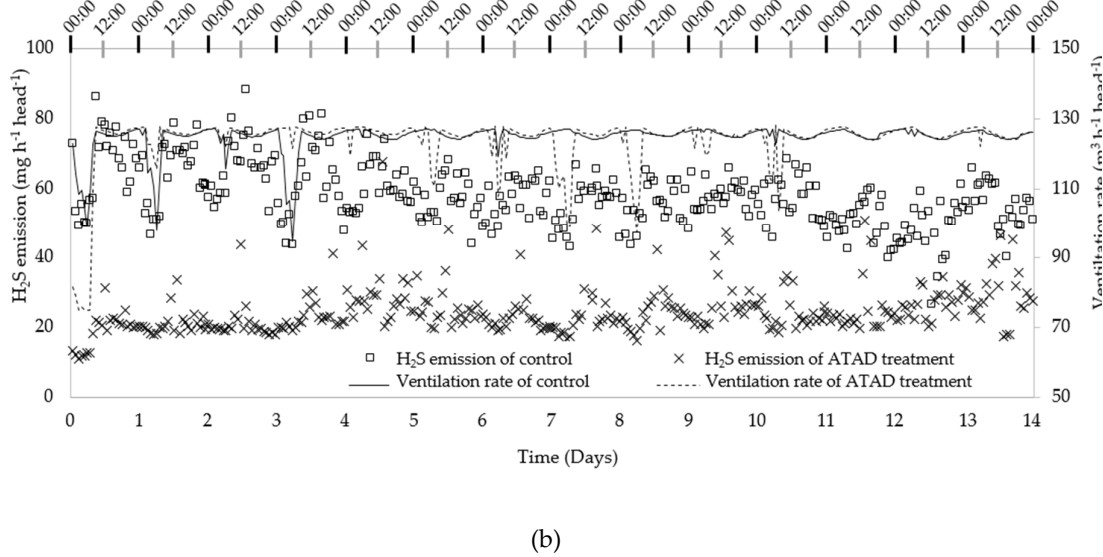

(b)

**Figure 6.** Comparison of hourly mean $H_2S$ concentrations and emissions for the control (conventional slurry pit) and ATAD treatment (pit recharge system) in summer: (**a**) variation of measured concentration (ppbv) and room temperature for each room (control and ATAD treatment); (**b**) estimated emission and ventilation rates.

The general trends for $H_2S$ emissions reflected the concentrations for each room, rather than the ventilation rates. The correlation coefficients (R) between $H_2S$ concentrations and emission rates were 0.97 and 0.99 for the control and ATAD treatment, respectively (Figure 7). Additionally, Figure A3 shows the correlations between ventilation rates at two other levels (low and maximum) and $H_2S$ emissions for the control and ATAD treatment. At low ventilation levels, the R-values between the ventilation rate and $H_2S$ emission were 0.36 and 0.89 for the control and ATAD treatment, respectively. At the maximum ventilation level, the ventilation rate and $H_2S$ emissions were poorly correlated with each other, i.e., R= −0.18 and −0.14 for the control and ATAD treatment, respectively.

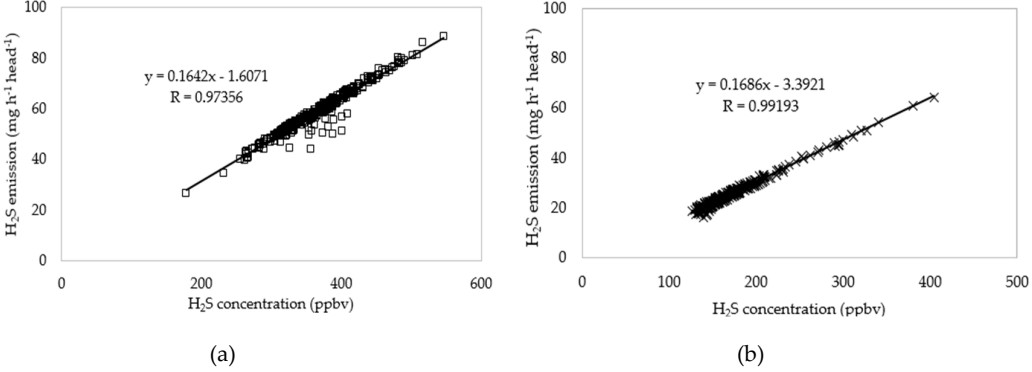

(a)                                        (b)

**Figure 7.** Correlation between $H_2S$ concentration and emission rates for the control (conventional slurry pit) and ATAD treatment (pit recharge system) in summer. (**a**) The correlation coefficient between the $H_2S$ concentration and emission for the control (conventional slurry pit) was 0.97, (**b**) while for the ATAD treatment (pit recharge system) was 0.99.

### 3.3. Daily Gas Concentrations and Emissions

The average daily mean ventilation rates, $NH_3$ and $H_2S$ concentrations, and emissions from each room are shown in Table 3. Due to the high outside temperature, the room temperatures for both rooms were not maintained at the set point temperature (25 °C), despite the maximum ventilation rates. The average room temperature for the control was about 31.7 °C, which was 0.5 °C higher than for the ATAD treatment during the experimental period ($p < 0.05$). The ventilation rates in both rooms were ~125 $m^3 \cdot h^{-1} \cdot head^{-1}$ ($p > 0.05$).

**Table 3.** Average of daily mean $NH_3$ and $H_2S$ concentrations and emission rates for the control (conventional slurry pit) and ATAD treatment (pit recharge system).

| | Control[1] | ATAD Treatment[2] | *p*-Value | Reduction Rate (%) |
|---|---|---|---|---|
| *n* | 14 | 14 | - | - |
| Room temperature (°C) | 31.7 ± 0.6 [a] | 31.2 ± 0.7 [b] | 0.0495 | - |
| Ventilation rate ($m^3 \cdot h^{-1} \cdot head^{-1}$) | 125 ± 1.8 [a] | 125 ± 1.5 [a] | 0.5820 | - |
| Gas concentration | | | | |
| $NH_3$ (ppmv) | 14.0 ± 0.9 [a] | 17.3 ± 0.7 [b] | <0.0001 | −24.4 ± 11.2 |
| $H_2S$ (ppbv) | 365 ± 35 [a] | 167 ± 17 [b] | <0.0001 | 53.7 ± 7.7 |
| Gas emission rate | | | | |
| $NH_3$ ($g \cdot d^{-1} \cdot head^{-1}$) | 25.9 ± 2.4 [a] | 33.7 ± 1.4 [b] | <0.0001 | −31.0 ± 14.4 |
| $H_2S$ ($mg \cdot d^{-1} \cdot head^{-1}$) | 1400 ± 132 [a] | 628 ± 47 [b] | <0.0001 | 54.6 ± 6.3 |

[1] Conventional slurry pit; [2] pit recharge system; [a, b] different superscripts in the same row, meaning each group is significantly different ($p < 0.05$).

Because of the relatively higher average $NH_3$ concentrations (by approximately 24%) in the ATAD treatment, the daily $NH_3$ emissions for the ATAD treatment were 33.7 $g \cdot d^{-1}$ $head^{-1}$, i.e., 31% greater than the control ($p < 0.05$, Table 3). On the other hand, the $H_2S$ concentration and emission rates for the ATAD treatment were significantly lower than for the control ($p < 0.05$). The reduction rates were 53.7 and 54.6% for the concentration and emission rates for $H_2S$, respectively, for the room equipped with a pit recharge system.

### 3.4. Characteristics of Recharging Liquid and Manure

The characteristics of the aerobically treated liquid manure and manure samples collected from each pit were analyzed for several parameters. As shown in Table 4, the aerobically treated liquid manure had high pH (8.6), which is generally an indicator of unstable manure. If the manure is stabilized through treatment in the ATAD system, the pH decreases due to the nitrogen fixed to the $NO_3^-$ form by nitrification [22,29]. Additionally, the $NH_4$-N content was 1860 $mg \cdot L^{-1}$, indicating that about 50% of the total N content of the aerobically treated liquid manure was in the form of the unstable $NH_4$.

**Table 4.** Characteristics of recharged aerobically treated liquid manure (collected from the last stage of the aerobic ATAD treatment system) and manure sample from the pits from the control and ATAD treatment. The samples were collected in the middle of the experiment (day 7). Manure samples from the pits represent stored manure.

| | Aerobically Treated Liquid Manure at Day 7 | Manure Sample from the Pits at Day 7 | |
| --- | --- | --- | --- |
| | | Control[1] | ATAD treatment[2] |
| Moisture contents (%, w. b.[3]) | 97.9 | 92.5 | 93.4 |
| Volatile solids (%, d. b.[4]) | 47.1 | 62.5 | 56.3 |
| pH | 8.6 | 7.8 | 7.9 |
| EC[5] ($\mu S \cdot cm^{-1}$) | 21.9 | 28.6 | 23.8 |
| Total N ($mg \cdot L^{-1}$) | 3580 | 7170 | 5630 |
| $NH_4$-N ($mg \cdot L^{-1}$) | 1860 | 4140 | 2720 |

[1] Manure from the control pit (conventional slurry pit); [2] aerobically treated liquid manure mixed with manure from the ATAD treatment pit (pit recharge system); [3] wet bases; [4] dry bases; [5] electric conductivity.

The manure sample from the control had a slightly lower moisture content than the manure from the ATAD treatment. The pH values for the manure samples from both rooms were similar, at 7.8 and 7.9 for the control and ATAD treatment, respectively; however, the EC, total N, and $NH_4$-N contents were higher in the control room.

## 4. Discussion

### 4.1. $NH_3$ and $H_2S$ Concentrations and Emission Rates in Summer Compared with Fall

This experiment was carried out in the same barn as the study by Wi et al. [22], and although the growth stage of pigs was similar, the $NH_3$ and $H_2S$ emissions in the control (conventional slurry pit) and ATAD treatment (pit recharge system) were remarkably different due to the direct and indirect effects of the seasons. Previous research reported that the pit recharge system could reduce the $NH_3$ concentration by 32.6%, but in summer (this research) the operation of the pit recharge system did not result in a reduction of the $NH_3$ concentration; in fact, the concentration increased by 24%. The averages of external temperatures were 29.0 and 17.3 °C in summer and fall, respectively (Table 5). Due to the differences in the external temperatures, the ventilation rates for fall were in the range of 47.0–62.0 $m^3 \cdot h^{-1} \cdot head^{-1}$, and averaged approximately 125 $m^3 \cdot h^{-1} \cdot head^{-1}$ in summer, which was 2–2.7 times higher than in fall. This difference contributed to higher $NH_3$ emissions. The higher ventilation rate in summer likely contributed to higher $NH_3$ emissions compared with fall. Although the ventilation rates in summer were maintained mostly at the highest level possible, the room temperature in summer was about 6.5 °C higher than in fall. This high temperature could be the reason for the increased $NH_4$-N concentration in the manure in the pit. Higher temperature likely activates urease in manure, which decomposes the organic N (urea) to $NH_4$-N. The higher temperature in summer also contributed to a higher gaseous $NH_3$ concentration by making it easier for the ammoniacal N in the manure to be released as gaseous $NH_3$ [23]. The increased N content in manure enhances the feasibility of $NH_3$ emissions from the manure. It is generally agreed that a 1% increase of the additional crude protein content could increase $NH_3$ emissions by approximately 20% [19]. The $NH_3$ concentration in the control room (conventional slurry pit) in summer (14.9 ppmv) was similar to fall (14.0 ppmv), despite the expected effect of dilution via higher ventilation. The $NH_3$ emission rate from the conventional slurry pit in summer was 1.9 times higher than in the fall (Table 5).

On the other hand, the mean $H_2S$ concentration in summer (0.4 ppmv) was lower than the fall concentration (1.1 ppmv). The mean $H_2S$ emission rate in summer was 8.7 $g \cdot d^{-1}$ $AU^{-1}$, ~65% of the

mean $H_2S$ emission rate in fall. However, the reduction rates for the $H_2S$ concentration (53.7%) and emissions (54.6%) in the pit recharge system were lower than in fall (where 78.3% and 83.7% reductions were observed for concentrations and emissions, respectively; Table 5).

**Table 5.** Comparison of gaseous $NH_3$ and $H_2S$ concentrations and normalized emissions for the animal unit (AU) in summer (this study) and fall [16].

| Seasons | Ambient Temperature (°C) | Room Temperature (°C) | Items | NH₃ | | | H₂S | | |
|---|---|---|---|---|---|---|---|---|---|
| | | | | Control [1] | ATAD Treatment [2] | Reduction Rate (%) | Control [1] | ATAD Treatment [2] | Reduction Rate (%) |
| Summer (This study) July | 29.0 | 31.5 | Concentration (ppmv) | 14.0 | 17.3 | −24.4[a] | 0.4 | 0.2 | 53.7[a] |
| | | | Emission (g·d⁻¹·AU⁻¹) [3] | 162 | 211 | −31.0[a] | 8.7 | 3.9 | 54.6[a] |
| | | | Ventilation rates (m³·h⁻¹·head⁻¹) | 125 | 125 | - | - | - | - |
| Fall (previous study [22]) October | 17.3 | 25.0 | Concentration (ppmv) | 14.9 | 10.3 | 32.6[b] | 1.1 | 0.2 | 78.3[b] |
| | | | Emission (g·d⁻¹·AU⁻¹) [3] | 86.3 | 41.5 | 53.3[b] | 13.4 | 2.1 | 83.7[b] |
| | | | Ventilation rates (m³·h⁻¹·head⁻¹) | 62.0 | 47.0 | - | - | - | - |

[1] Conventional slurry pit; [2] pit recharge system; [3] daily gas emissions normalized for 500 kg of live animal weight; [a, b] different superscripts in the same column, meaning each item in different seasons is significantly different ($p < 0.05$).

*4.2. Correlation between Ventilation Rates and Gas Emissions*

The correlations between the ventilation rates and gas emissions at low ventilation levels were higher than that for the maximum level of ventilation (Figures A2 and A3) for both $NH_3$ and $H_2S$. The reason for analyzing correlations between the ventilation rates and gas emissions at two ventilation levels (low and maximum) was based on the close inspection of the data in Figures 4 and 6, and consideration of the two-film theory for mass transfer [30]. For low ventilation levels, the impact of the increasing ventilation rate is apparent (i.e., a decreasing thickness of the boundary layer increases the stripping of the $NH_3$ and $H_2S$ gases from the surface of the manure). However, for high ventilation levels (for which the boundary layer is already minimized), the amount of stripped gas is no longer correlated with ventilation rates (Figures A2 and A3).

*4.3. Seasonal Effect on Characteristics of Recharging Liquid and Manure*

The aerobically treated liquid manure differed between summer and fall [22] in several parameters (Table 6). Although the pH difference between two seasons (pH 8.6 and 8.4 in summer and fall, respectively) was relatively small, a clear seasonal difference in EC was observed. The EC value of the aerobically treated liquid manure was also high (21.9 $\mu S \cdot cm^{-1}$), which was in the reported range of EC for raw swine manure (12–24 $\mu S \cdot cm^{-1}$) [31]. The total N and $NH_4$-N contents in the recharging liquid were 1.9 and 2.1 times greater than in the fall, respectively. Additionally, the total N content of the manure sample from the ATAD treatment pit in summer was 5630 mg $L^{-1}$, which was ~5 times higher than the N content in the fall season manure. Additionally, 2720 mg $L^{-1}$ of $NH_4$-N content in the manure in summer contributed to the ~5 times greater $NH_3$ emissions compared with the fall.

**Table 6.** Comparison of aerobically treated liquid manure and manure sample from the ATAD treatment pit in summer (this research) and fall [22].

| | Summer (this study) | | Fall [22] | |
|---|---|---|---|---|
| | **Aerobically Treated Liquid Manure[1]** | **Manure Sample from ATAD Treatment[2]** | **Aerobically Treated Liquid Manure[1]** | **Manure Sample from ATAD Treatment[2]** |
| Moisture contents (%, w.b.[3]) | 97.9 | 93.4 | 98.7 | 98.3 |
| Volatile solids (%, d. b.[4]) | 47.1 | 56.3 | 40.1 | 45.7 |
| pH | 8.6 | 7.9 | 8.4 | 8.2 |
| EC [5] ($\mu S \cdot cm^{-1}$) | 21.9 | 23.8 | 12.9 | 12.7 |
| Total N (mg·$L^{-1}$) | 3580 | 5630 | 1190 | 1130 |
| $NH_4$-N (mg·$L^{-1}$) | 1860 | 2720 | 567 | 633 |

[1] Recharging liquid for the treatment pit sampled from the last stage of the ATAD system; [2] aerobically treated liquid manure mixed with manure from the ATAD treatment pit (pit recharge system), sampled at the middle depth of the pit (Figure 1) on days 7 and 13 of each experimental period for summer and fall [22], respectively; [3] wet basis; [4] dry basis; [5] electric conductivity.

The high N contents in manure samples collected in summer can be explained by the feeding program in hot seasons and annual management of the ATAD systems in swine farms. The feeding program in summer for the finishing pigs uses high crude protein rations (used by farmers to overcome heat stress and improve weight gain). When compared with the fall feed, the crude protein content was 1% higher in summer (Table 1). Feeding finishing pigs with protein-rich feed can cause

more excretion of undigested N as manure. The increased N content in the manure enhances the feasibility of the $NH_3$ emissions from the manure. It is generally agreed that a 1% increase of additional crude protein content could increase $NH_3$ emissions by approximately 20% [19]. High N content in manure can induce increased N influx to the ATAD system. This causes N overload of the ATAD system, which in turn flushes the recharging liquid with high $NH_4$-N concentration into the pit. Other feed ingredients (such as fermentable carbohydrates) may also influence $NH_3$ emissions [32]. The research on seasonal effects and feed rations is warranted.

The annual practice of managing the ATAD system can affect the N characteristics of the recharging liquid. Inadequate operation of ATAD system in spring and fall disrupting the microbial balance in the ATAD system. An ATAD system with a disturbed microbial balance produces aerobically treated manure with high $NH_4$-N content. This unstabilized recharging liquid can affect gaseous emissions from the recharged pit. The pumping out of the treated manure occurred approximately two months before the experimental period. However, the ATAD system was likely affected by the long recovery time needed for the microbial balance. These indirect effects of summer on the recharging liquid caused more $NH_3$ emissions in the treatment (pit recharge system) than in the control (conventional slurry pit).

## 5. Conclusions

The effects of recharging (manure) liquid on $NH_3$ and $H_2S$ emissions from a commercial swine farm equipped with a semicontinuous pit recharge system were evaluated over 14 days in summer. Pigs were fed summertime feeding rations (used by farmers to cope with animal heat stress), and the pit manure properties were also affected by temperature and management. Gas concentrations and emissions from a room equipped with a pit recharge system were compared with those from a room operating a conventional slurry pit under a fully-slatted floor. The $NH_3$ emissions were 31 ± 14% higher ($p < 0.0001$) and the mean reduction of $H_2S$ emissions were 55 ± 6% ($p < 0.0001$) in pit recharge system room. The use of feed with high crude protein content, the high temperature of the manure surface, and increased ventilation rates contributed to high $NH_3$ emissions in summer (~2 times higher than in the fall). In addition, the annual practice of pumping out the aerobically treated liquid manure from the on-farm storage tank and part of the ATAD system caused the pit to be recharged with high $NH_4$-N containing liquid, thereby contributing to increased $NH_3$ emissions. It is recommended that completely stabilized recharging liquid be used in ATAD for the pit recharge system. Future research will need to measure and control ATAD operation parameters (i.e., HRT, temperature, etc.). Additionally, other research showed that low crude protein content in feed reduced $NH_3$ emissions, suggesting that testing of other feeding methods for summer is still warranted. Some countries regulate maximum emission levels from livestock farming. The results of this research provide farm-scale data about baseline (summertime) emissions and the effectiveness of the mitigation of gaseous emissions, which can be used in the portfolio of technologies that are available to the livestock industry.

**Author Contributions:** conceptualization, H.A.; methodology, H.A.; validation, J.W., J.K., and H.A.; formal analysis, J.W.; investigation, J.W., S.L., E.K., and M.L.; resources, S.L., E.K., M.L., and H.A.; data curation, J.W.; writing—original draft preparation, J.W.; writing—review and editing, J.W., J.K, and H.A.; visualization, J.W.; supervision, H.A., and J.K.; project administration, H.A.; funding acquisition, H.A. All authors have read and agreed to the published version of the manuscript.

**Funding:** This research was supported by the Rural Development Administration (Project No. PJ 01385001), Republic of Korea. This project was partially supported by the Iowa Agriculture and Home Economics Experiment Station, Ames, Iowa. Project no. IOW05556 (Future Challenges in Animal Production Systems: Seeking Solutions through Focused Facilitation), which is sponsored by the Hatch Act and State of Iowa funds.

**Acknowledgments:** This authors gratefully acknowledge Woosang Lee (Smart Control and Sensing, Inc.) for his help with gas monitoring.

**Conflicts of Interest:** The authors declare no conflict of interest. The funders had no role in the design of the study; in the collection, analyses, or interpretation of data; in the writing of the manuscript, or in the decision to publish the results.

## Appendix A

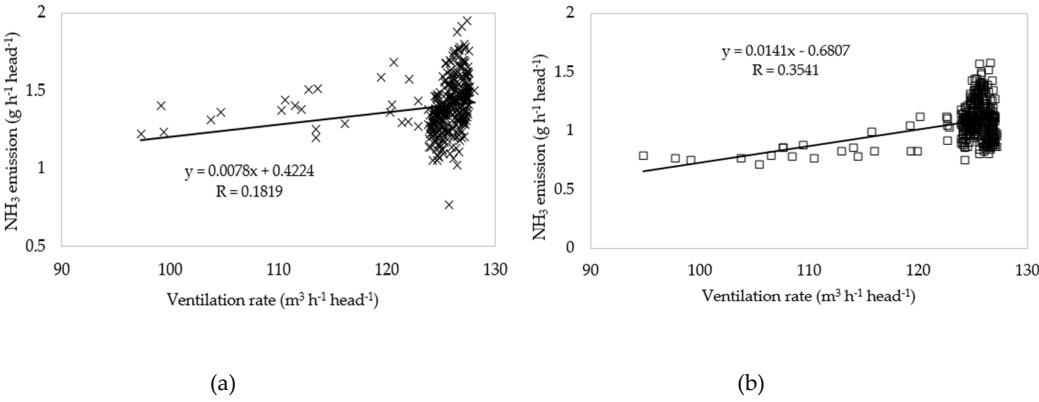

(a)                                                                                              (b)

**Figure A1.** Correlation between the ventilation rate and $NH_3$ emissions in the (**a**) control (conventional slurry pit) and (**b**) treatment (pit recharge system) systems.

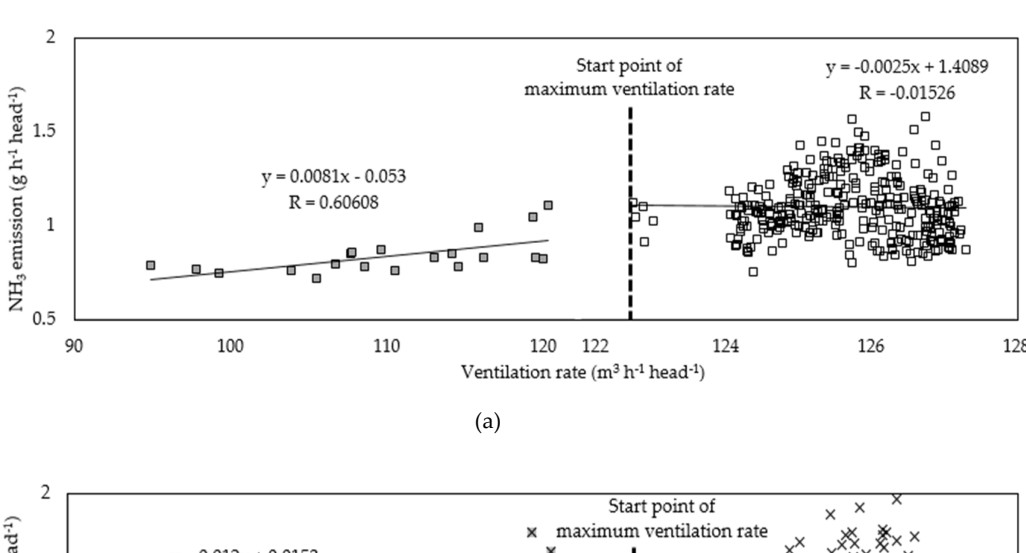

(a)

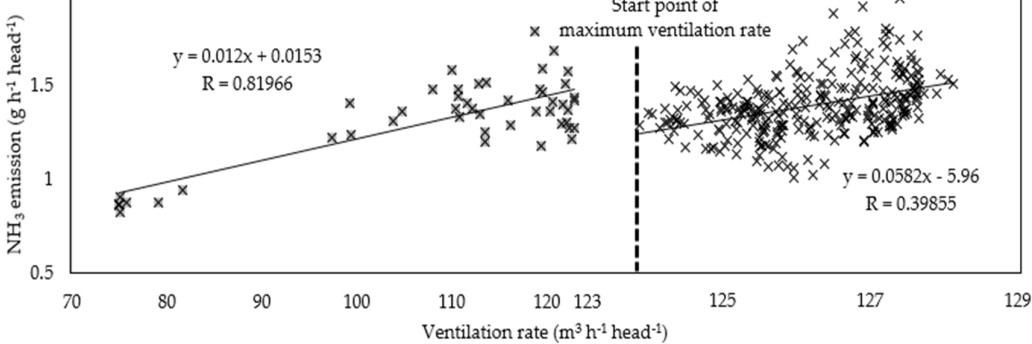

(b)

**Figure A2.** Correlation between the ventilation rate and $NH_3$ emissions in the (a) control (conventional slurry pit) and (b) treatment (pit recharge system) systems. All values are divided into two ventilation rate levels—low and maximum.

**Table A1.** Ventilation range and average room temperature for each ventilation level (low and maximum). The start point of the maximum ventilation rate was determined by the measurement of the operating rate of the fans.

|  | Low Ventilation Level | | Maximum Ventilation Level | |
|---|---|---|---|---|
|  | **Ventilation Range ($m^3\ h^{-1}\ head^{-1}$)** | **Room Temperature (°C)** | **Ventilation Range ($m^3\ h^{-1}\ head^{-1}$)** | **Room Temperature (°C)** |
| Control | 94.9 ~ 120.3 | 27.8 ± 0.5 | 122.7 ~ 127.3 | 31.9 ± 2.3 |
| Treatment | 75.0 ~ 122.8 | 28.0 ± 0.9 | 123.9 ~ 128.1 | 31.6 ± 2.7 |

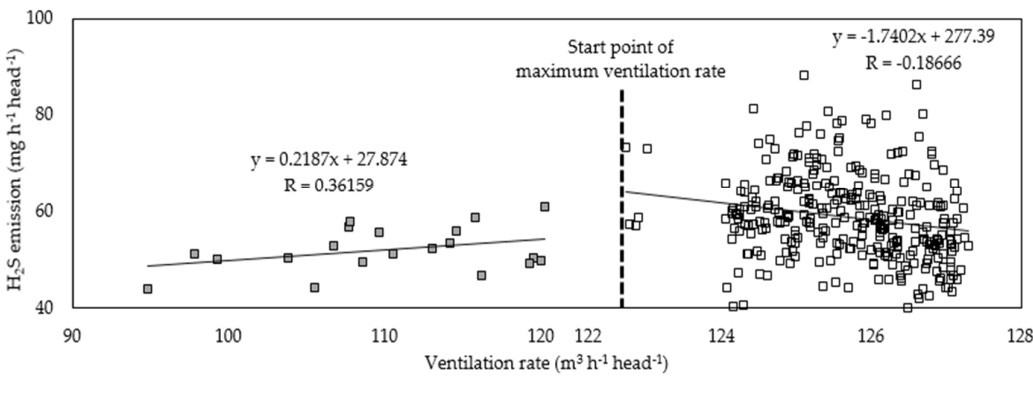

(a)

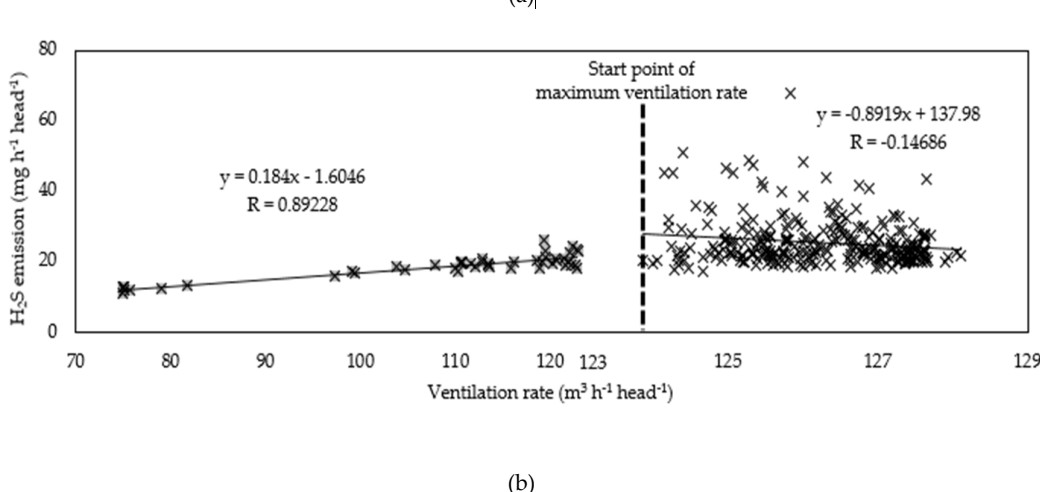

(b)

**Figure A3.** Correlation between ventilation rate and $H_2S$ emissions in the (**a**) control (conventional slurry pit) and (**b**) treatment (pit recharge system) systems. All values divided into two ventilation rate levels—low and maximum.

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
