# Peer review of "Effects of Treated Manure Conditions on Ammonia and Hydrogen Sulfide Emissions from a Swine Finishing Barn Equipped with Semicontinuous Pit Recharge System in Summer"

_atmosphere, doi:10.3390/atmos11070713_

Round 1
Reviewer 1 Report
I believe that this study is almost a follow-up from previous research (Wi et al. 2019) published in this Journal. The authors looked for evaluating the summer behavior for both pit systems, but the most of the text is focused in previous results (fall evaluation). I believe that this paper inside the scope of Atmosphere, but must be improve. I recommend that authors will do statistical comparison between summer and fall emissions for strengthening the paper.
- General Comments
- The title is not clear. First sentence “Effect of treated manure characteristics…” must be modify. The word: characteristics could be confused for reader. Moreover, in the title the most important is the comparison between semi-continuous pit recharge system and control treatment, but in the discussion is focused in emissions between summer and fall.
- Sometimes the authors used NH3 and others ammonia, or H2S and Hydrogen sulfides, those must be standardized.
- The authors decided to write results and discussion separately, so the results section only must be included the results in a concise and precise manner. Hence, in this section must not contain references to other works.
- Introduction
- Lines 44-55: The authors give us information about the importance for studying NH3 and H2S emissions from livestock facilities. They mentioned H2S effect over animals, but what is the effect of NH3 over animals? This is important because the authors said that this study give a unique approach to mitigate emissions and improve air quality inside.
- Lines 44-55: Is there some air quality norm or important papers over emissions in livestock facilities? This information could improve this study and give it more strength.
- Lines 72-73: The sentences “… farmers feed a high-crude protein diet to overcome heat stress and improve weight gains” must be referenced.
- Experiments
- Table 1. The authors give us the composition of swine food for this research and previous study [16], but the percentage of feed in “Feed A” is 26.21% and in “Feed B” is 26.14%. I would like to know what are the others parts or ingredients in each feed. How could those ingredients affect the emissions?
- Line 159. Please, to indicate the number or reference to APHA methods employed. This give the possibility to replicate the assays.
- Lines 170-173. How many replicates or repeats were evaluated in this statistical analysis??
- Results
- Line 176. The authors said: “The hourly trends… Figure 4”, but if I saw the figure 4 I cannot see the hourly, I only saw “days” as measure unit no “hours”. Furthermore, I did not read in experiment section when NH3 or H2S measure began… So, I believe that information about hourly must be added.
- Lines 178-180: I recommend that statistical comparison between both rooms should be mentioned here, because in this way is easier for reader to understand the differences between two treatments.
- Lines 180-183: Wi et al. [16] must be moved to discussion.
- Line 192. I had the same problem that line 176 when the authors said “except for the early morning…” I did not know when is early morning in this experiment.
- Figure 4b. After 3th day the behavior of ventilation rate of control treatment was different to ventilation rate of treatment. In the former was almost a continue line while in the latter had many under peaks… How can this affect the NH3 emissions? Why did happen this?
- Lines 196-197. Where can I see these correlations? In Figure A1 this is not possible to observe.
- Lines 202-205. When did maximum ventilation level is necessary? How was this determine? Is relevant low ventilation rate for this study, if I can see very few dates in comparison with maximum ventilation rate?
- Lines 205-206. Why was different the time for maximum ventilation rate between both rooms if they are identical?
- Lines 212-217. Discussion
- Lines 227-229. Same comment that 4.b.
- Figure 6b. Same comment that 4.e.
- Lines 243-247. Discussion. Moreover, it was explained the same, y with the same words, that in lines 214-217. The authors must will add in discussion one section or paragraph with that information.
- Lines 258-259. The authors declared that temperature showed a significant difference between control and treatment rooms, but in Table 3 both temperatures have the same letter (a) and the p-value is almost 0.05. I’m not sure if this is statistically different.
- Lines 265-269. Add table 3 for cross reference.
- Lines 274-275. Discussion.
- Table 4. Is there statistical analysis for this information? I believe that is necessary.
- Discussion
- Line 292. The authors highlighted the word remarkably… I would like to know how they determined that this difference is remarkably. I believe that could be very good that authors will make a statistical analysis included information from both seasons, especially because all discussion include information from Wi et al. [16] and this was a study from the same authors, as they declared.
- Lines 299-300. Reference.
- Lines 301-304. Add Table 5 at the end of sentences.
- Lines 305-309. Same comment as above.
- Table 5. I recommend to realize statistical analysis between two seasons.
- Table 6. I did not find the information of “Manure sample from treatment” from [16]. Please check that information.
- Conclusions
- Lines 346-348. The sentence is unfinished. Anyway, I don’t know if it gives useful information.
- Lines 353-355. The authors mentioned that “…increased ventilation rates contributed to high NH3 emissions” I’m not sure about this due to that in Figure A1a the correlation between ventilation rate and NH3 emissions was near to 0. Please clarify this.
- Lines 353-355. I did not find inside the text that higher NH3 emissions in summer that in fall were due to the ventilation rate. More information or analysis are necessary for this declaration. I could assume that the crude protein is the most important for increase NH3 emissions because there is previous information that indicated this [19]
- Lines 358-359. I find that is very complicated recommended to use low crude protein content feed is required because the farmers have reasons for doing this diet change. Perhaps, adding a recommendation for testing different diet may be appropriate.
Reviewer 2 Report
Nice job! It was a really interesting study in conjunction with the previous study to not only quantify the effect of the system but also compare it in a different season with changes in food characteristics.
- Line 195-197: Correlation between ventilation rate and ammonia emission was discussed while correlation coefficient (r= 0.35 and 0.18) is not presented in any chart/table. This correlation again is discussed in the next paragraph. Please consider combining this with the next paragraph.
- Line 201-202: “every value was divided by two levels of ventilation rate, low and maximum.” It seems confusing to me so I am suggesting: ventilation rates were grouped in two levels, low and maximum.
- Line 314-315: I am not sure if we can conclusively talk about pH changes in the system since it only increased 0.2 units in ATAD while it decreased 0.3 units in pits which play the key role in emission. In addition, evaluation of the difference between them is not possible statistically with one sample.
- Line 332-333: While, based on table , crude protein increased ~1%-point from fall to summer (17.48% vs. 18.36%), ammonia emission increased 88% in control room (86.3 vs 162 g/d/AU) and 408% in treatment room (41.5 vs 211 g/d/AU). Please consider mentioning/justifying this as 1%-point crude protein increased ammonia emission considerably.
- Table 6: Considering that pit was recharged three times a day, It would be beneficial if you could describe why, in the fall, Total N remained almost constant while it drastically decreased in ATAD in summer.
- In the introduction (line 69) operational condition of ATAD was mentioned as influencing parameter in recharging liquid characteristics. However, these conditions were not provided, specifically recharge temperature and HRT. Also. Please provide temperature of manure in the control pit if it is available. If not, suggest it be measured in future research since circulation manure outside of pit could increase temperature in summer as it heated by the sun. This could also justify higher mineralization of organic nitrogen in the treatment room due to higher temperatures.
- In results and discussion, higher emission of ammonia in summer compared to fall has been discussed. However, possible causes of higher emission of ammonia from treatment compared to control in summer itself were not discussed. It could be higher manure temperature in treatment pit since it is being circulated outside and heated by the sun. Also, oxygen solubility in water and oxygen transfer to manure/water in ATAD decreases with increasing temperature in summer. Thus, please consider a short discussion specifically for summer, possibly in section 3.3, as to why ATAD result in higher ammonia emission.
Reviewer 3 Report
This study describes NH3 and H2S concentrations and fluxes from a swine houses employing an ATAD treatment and compares it to a swine house using a standard pit accumulation system. The methods are well described and the data presented present an interesting case for carefully using the ATAD system during summer with NH3 emissions actually increased while H2S decreased. The English language usage in the abstract and introduction was quite good, but the quality decreased dramatically in the results and discussion sections and needs to be improved. Lastly, a few general suggestions--i) throughout the manuscript instead of just 'treatment' use 'ATAD treatment' or something similar, and ii) be careful to make a distinction between dissolved ammonium,gaseous NH3 concentration, and NH3 flux. Too often you talk about NH3, and the reader can be easily confused about what NH3 specie you are describing/comparing.
Finally, I wonder if part of the greater NH3 flux may be related to hot animals in summer. Greater water intake, more N as urea (instead of complex organic N in feces), urea is quickly converted to NH3 in microbially enriched ATAD pits?
Specific comments:
L123: continuous mode.
L133: introduced into the growing
L137: pigs were fed
L139: Table 1 describes
L144: with cool-season diet
L159: pH and EC were measured
L177: rooms are plotted in Figure
L200: Delete 'In addition,' change emission is depicted
L240: shows the correlation
Table 3: Shouldn't the Room Temp superscripts be different? NH3 p-value should read '<0.0001'
L265: higher average
L272: pit were analyzed for several
L273: value, indicating Question: Why does high pH indicate unstabilized manure? Maybe cite a reference or report the pH range for stabilized manure.
L285: had slightly lower
L300: organic N to NH4-N. The higher temperature in summer also contributed to higher gaseous NH3
L310: Comparison of gaseous NH3
L313: manure differed between summer and fall
L316: from the ATAD treatment pit in summer [This is a good example where 'pit of treatment' is better described as an ATAD treatment]
L329: When compared with fall feed, the crude protein content was 1% higher in summer (Table 1).
L335: system and flushes recharging liquid
L336-344 and Conclusions: Needs to be written more clearly. Several subject-verb issues and awkwardly written sentences.
L370: Facilitation) which is sponsored
Round 2
Reviewer 1 Report
The manuscript has made considerable progress toward a publishable form, but there is some questions that must be answered before it publication in Atmosphere.
General Comments
- According to authors’ answer I understood that there is not a norm about emissions in livestock facilities. I suggest them including in discussion about the necessity about legislate or establish permissible maximum limits in livestock facilities.
- In my previous review, I had a doubt about the other ingredients present in composition of feed in Table 1. The authors responded that those ingredients are minerals and nitrogen free extract (as glucose or sugar), but they don’t answer about the effects of those ingredients over NH3 or H2S emissions. For instance: Le et al. (2008) said that to high level of fermentable carbohydrates reduced NH3 So, I would like that the authors would discuss or, at least, indicate their influence in the emissions.
- In my previous review, I asked the authors information about Figure 4b, and as the difference to ventilation rate can affect the NH3 emission when control treatment is maintained constantly vs ATAD treatment where there is a negative peak during many hours. I can understand the authors’ explication, but they didn’t deepen about the effect between to increase the ventilation rate and NH3 emissions. Perhaps this didn’t have effect over NH3 emission but I believe that is necessary an explication about this, especially for readers without the specific knowledge, as me.
- In the previous review, two reviewers had the same doubt about the correlation coefficients (R=0.35 and 0.18), because there is no reference in the manuscript where we can observe this. The authors modified the manuscript but they didn’t indicate where we can see this correlation (again). On the other hand, the authors said (in theirs answers) that in Figure A1 we can see this information, but in figure the correlation coefficients indicated (R=0.60 and -0.015) are different. Please, clarify this, maybe those Rs (0.35 and 0.18) were obtained with all values without separation between the two ventilation levels.
- The authors in theirs answer said that from table 4 only take data on Day 7th, but I would like if the authors take one sample that day or more than one.
- In the previous review, Reviewer 2 requested an explication or justification about as 1%-point crude protein increased ammonia emission considerably (88% and 408%). In the manuscript the authors only indicated that 1%-point more protein could increase 20% NH3 Therefore, this answer must be addressed in the manuscript.
Line 51. Change to subscript PM2.5
Line 105. Change to Superscript 17th to 30th
Line 129. Modify Figure for Figures
Lines 135-136. I suggest changing the sentences: “…are presented in Figures 1, A1, A3 in the previous manuscript [22]”, for “…are presented by Wi et al. [22] (Figures 1, A1 and A3)”
Line 147. After Feed A cited the Table 1. i.e. “Feed A (Table 1)”
Line 148. Change is for was in “…weight of pig is similar to this research”
Line 226. Include (Figure A1) after respectively.
Line 233. Include (Figure A1) at the end of the sentence.
Line 297. Change & for and.
Lines 303-306. Discussion
Line 332. Change higher in the second high
Line 417. Delete references from conclusion
References:
Le et al. (2008). Interactive effects of dietary crude protein and fermentable carbohydrate levels on odour from pig manure. Livestock Science 114: 48-61.
